# Biogeographic Patterns of Fungal Sub-Communities under Different Land-Use Types in Subtropical China

**DOI:** 10.3390/jof9060646

**Published:** 2023-06-06

**Authors:** Hao Liu, Heming Han, Ruoling Zhang, Weidong Xu, Yuwei Wang, Bo Zhang, Yifan Yin, Hui Cao

**Affiliations:** Key Laboratory of Agricultural Environmental Microbiology, Ministry of Agriculture and Rural Affairs, College of Life Sciences, Nanjing Agricultural University, Nanjing 210095, China; 2022216010@stu.njau.edu.cn (H.L.);

**Keywords:** land management practices, fungal, sub-communities, distance–decay relationship, assembly processes

## Abstract

Revealing the regional distribution and diversity of fungal sub-communities under different land management practices is essential to conserve biodiversity and predict microbial change trends. In this study, a total of 19 tilled and 25 untilled soil samples across different land-use types were collected from subtropical China to investigate the differences between the spatial distribution patterns, diversity, and community assembly of fungal sub-communities using high-throughput sequencing technology. Our results found that anthropogenic disturbances significantly reduced the diversity of abundant taxa but significantly increased the diversity of rare taxa, suggesting that the small-scale intensive management of land by individual farmers is beneficial for fungal diversity, especially for the conservation of rare taxa. Abundant, intermediate, and rare fungal sub-communities were significantly different in tilled and untilled soils. Anthropogenic disturbances both enhanced the homogenization of fungal communities and decreased the spatial-distance–decay relationship of fungal sub-communities in tilled soils. Based on the null model approach, the changes in the assembly processes of the fungal sub-communities in tilled soils were found to shift consistently to stochastic processes, possibly as a result of the significant changes in the diversity of those fungal sub-communities and associated ecological niches in different land-use types. Our results provide support for the theoretical contention that fungal sub-communities are changed by different land management practices and open the way to the possibility of predicting those changes.

## 1. Introduction

Microorganisms are critical for maintaining the functionality of soil ecosystems [1,2], and thus agricultural productivity depends on having diverse microbial communities in soil [3]. As demand for food has grown rapidly, in line with the increases in the global population, however, land in its natural environment, such as forests, grasslands, and wastelands, is often transformed into arable land for crop production [4]. Such agricultural land is generally under high-intensity utilization compared to natural land and is subjected to fertilization, irrigation, pesticides, and other agricultural management practices that affect microbial communities [5].

Anthropogenic disturbances can in particular alter soil environmental factors, such as soil pH and nutrients, and thus affect the microbiome of terrestrial ecosystems [6,7]. Soil microbiomes converge toward similar microbial properties to improve their adaptability in the same soil environment. It is possible that intensive modern agricultural management practices (e.g., fertilization) may encourage the spatial homogenization of microbial communities [8]. It has been shown, for example, that the conversion of forests to agricultural land results in more homogenous microbial communities and a decrease in microbial diversity [9]. A study on grasslands also found that a moderate increase in land-use intensity also led to the biohomogenization of microbial communities above and below ground [10]. Given the role of microorganisms as mediators of many important ecosystem functions, the loss of microbial diversity during ecosystem conversion should be of concern [11]. Although comparative studies of the relationship between land management practices and changes in microbial communities have been reported, it remains uncertain whether there are significant differences in community diversity traits between different fungal sub-communities at a regional scale.

Microbial communities often exhibit skewed abundance distributions, including a few abundant taxa with a wide distribution and numerous rare taxa with a limited distribution [12]. Traditionally, studies of the ecological functions of microbial communities have tended to focus on the abundant sub-communities, while neglecting the rarer sub-communities [13]. Recent studies, however, have shown that rare fungal taxa, which likewise include diverse genes and functions [14], can serve as an equally valuable genetic repository compared to the more abundant and metabolically active strains, and these rare taxa also play an important role in community function and stability [13]. Furthermore, abundant and rare sub-communities that have different ecological strategies [15] mean that the structure of those sub-communities is influenced by different controlling factors or ecological processes [16,17]. Compared to abundant taxa, rare taxa have a narrower ecological niche width and lower competitive ability, and are more influenced by external factors, while abundant taxa tend to have a wider ecological niche width and higher competitive ability and are less influenced by external factors [15]. It has therefore been suggested that the construction of abundant fungal sub-communities is mainly controlled by stochastic processes, while the construction of rare fungal sub-communities is mainly controlled by deterministic processes [16]. While these studies tend to suggest that fungal sub-communities may exhibit different responses to different land-use practices, they tend to conflate the large number of species in the community, thus ignoring differences between abundant and rare taxa, and few have examined changes in the response of different fungal sub-communities under different land management practices.

In this study, we conducted a regional survey of fungal communities by collecting forty-four soil samples (including nineteen tilled and twenty-five untilled samples) from Xuancheng, China. Using high-throughput sequencing technology, the response of soil fungal sub-communities to anthropogenic disturbances was investigated, as well as the impacts on community construction. We hypothesized that: (1) the responses of fungal sub-communities to different land-use practices may vary inconsistently due to differing adaptive capacities; (2) fungal sub-communities become more homogeneous after anthropogenic disturbances; and (3) changes in land-use patterns alter the balance between deterministic and stochastic processes with respect to the assembly of fungal sub-communities.

## 2. Materials and Methods

### 2.1. Study Site and Soil Sampling

Our study was conducted in Xuancheng City (between 29°57′–31°19′ N and 117°58′–119°40′ E) Anhui Province, China. The sampling sites are subject to a subtropical monsoon climate with a multi-year mean temperature of 16.0 °C. Xuancheng City has a total land area of 1,235,566 hectares, of which 215,868 hectares are cultivated, accounting for 17.47% of the total land area. The predominant soil types here are sandy and clay soils. The vegetation resources are abundant, and the common vegetation is mainly evergreen broad-leaved forest, deciduous broad-leaved mixed forest, or coniferous broad-leaved mixed forest. The main crops planted are rice and wheat. Xuancheng city’s natural conditions, including climate, topography, landforms, farming systems, and socioeconomic conditions including productivity, are typical of the cities represented in southern China.

Soil samples were collected in October 2019. A total of 44 sites were sampled, of which 19 were from tilled land (vegetable gardens, paddy fields, farms, and orchards) and 25 were from untilled land (woodland, wasteland, dryland, and grassland). According to the survey, the tilled soils had more than 2 years of cultivation history and the untilled soils had a minimum of 10 years without anthropogenic disturbance (Appendix A). Soil cores (6 cm diameter) without plant roots were randomly collected from five points with a soil driller and divided into two depth layers: 0–20 cm and 20–40 cm (including: Tillage topsoil layer, TT; Tillage subsoil layer, ST; No-tillage topsoil layer, TN; No-tillage subsoil layer, SN), for a total of 88 soil samples. Fresh samples collected in the field were immediately stored in sterile bags and transported to the laboratory under cold conditions. The samples were sieved in a 4 mm aperture sieve in the laboratory to remove roots and rocks. Samples from each sampling site were taken at depth, homogenized after sampling, and then divided into two parts for subsequent analysis. The soil samples that were to be analyzed for their physical and chemical properties were air-dried and sieved with a 2 mm sieve, while the samples intended for DNA extraction were stored at −80 °C.

### 2.2. Determination of Soil Physicochemical Properties

The soil pH, organic matter (SOM), total nitrogen (TN), total phosphorus (TP), available phosphorus (AP), total potassium (TK), and available potassium (AK) were measured as previously described [17,18].

### 2.3. Soil DNA Extraction and 16S rRNA Gene Sequencing

A FastDNATM Spin Kit (MP Biomedicals, Santa Ana, CA, USA) was used to extract DNA from 0.5 g of soil (fresh weight) following the manufacturer’s instructions. The extracted DNA was then purified using the PowerClean DNA Purification Kit (Mo Bio, Carlsbad, CA, USA), and its concentration and quality measured using a NanoDrop ND-1000 spectrophotometer (Thermo Scientific, Wilmington, DE, USA). The fungal ITS1 and ITS2 regions were amplified using 20–50 ng of DNA as a template, using the upstream primer containing the GTGAATCATCGARTC sequence and the downstream primer containing the TCCTCCGCTTATTGAT sequence. The PCR reaction parameters across a total of 24 cycles were: pre-denaturation parameter 94 °C, 3 min; denaturation parameter 94 °C, 5 s; annealing parameter 57 °C, 90 s; extension parameter 72 °C, 10 s; final extension parameter 72 °C, 5 min. After the amplification products were detected, PE250 paired-end sequencing was performed with an Illumina MiSeq (Illumina, San Diego, CA, USA) instrument to obtain the raw sequencing data.

### 2.4. Processing of Sequence Analysis

Unique barcodes were assigned to the paired-end reads, and quality filtering was carried out by Cutadapt (V1.9.1) to obtain high-quality and clean reads [19]. Reads that were less than 200 bp in length and with an average base quality score <20 were considered as low-quality sequences and omitted from further analysis. The reads were validated using the UCHIME algorithm in the Silva database with chimeric sequences being classified and eliminated [20]. Operational taxonomic units (OTUs) were constructed from sequences with at least 97% similarity. Annotated analysis of OTUs was performed with the Unite Database (https://unite.ut.ee/, accessed on 23 September 2022). using the Mothur algorithm to retrieve representatives of taxa and to calculate the number of OTUs for each sample in the taxonomic information [21]. MEGA-X (V11.0) was used for phylogenetic analysis [22]. Genomic sequencing data used in this study were submitted to the NCBI Sequence Read Archive (SRA) under the accession number SRR24744258 to SRR24744345.

### 2.5. Statistical and Bioinformatic Analyses

We followed previous studies in defining rare, intermediate, and abundant fungal sub-communities as a basis to investigate the variation in fungal communities under different management practices [23,24]. A relative abundance of OTUs above 0.1% was defined as “abundant,” while OTUs below 0.01% were defined as “rare” and OTUs with relative abundances between 0.1 and 0.01% were defined as “intermediate.” In total, in tilled and untilled samples, respectively, 4.01% and 5.53% of OTUs were classified as abundant, 22.20% and 29.22% were classified as intermediate, and 73.79% and 65.25% were classified as rare. Meanwhile, abundant taxa showed a significantly higher proportion in relative abundance (mean = 78.04%), compared with other taxa (Appendix A). Statistical analysis, however, found no significant difference in community composition between the top and subsoil layers of either the tilled or the untilled soil samples (Appendix A), and dividing the data into four groups (i.e., TT, ST, TN, and SN) was not a valid grouping (*p* > 0.05). The soil layers were therefore combined for the purposes of this study and no distinction was made between soil layers; instead, only tillage (TT) and non-tillage (NT) treatments were distinguished.

All analyses in this study were performed through various libraries/packages in R studio (running R version 3.6.1). To assess the alpha diversity of fungal communities, the Chao1 richness and Shannon diversity indexes were calculated using the “Vegan” package (version 2.5-5) in R, while non-parametric Wilcoxon rank sum tests were used for analysis of variance [25,26]. The principal coordinate analysis (PCoA) based on the Bray–Curtis distance was used for ranking analysis to reveal compositional differences for β-diversity in fungal communities [27]. The validity of the groupings was assessed through a PERMANOVA analysis with 9999 permutations, performed using the “Adonis” function in the R package “Vegan” [28]. The homogeneity of fungal communities across treatments was calculated using the “betadisper” and “permutest” functions of the “Vegan” package [29].

Niche width refers to the sum of the different resources utilized by a population in a community [30]. The wider the niche of a species, the less specialized the species is and its preference is to be a generalist species, which has a strong competitive ability. The narrower the niche of a species, the more specialized the species is and its preference is to be a specialist species, which is generally at a disadvantage in resource competition [31]. In this study, we followed Wu et al.’s [32] approach to assess niche breadth, calculating the ecological niche width of species using Levins’ niche breadth index, and then followed Xiong et al.’s [33] approach, where species within the community with a niche width greater than 3 were considered as generalists and species with a niche width less than 1.5 were considered as specialists.

The distance decay rate of fungal communities was obtained by calculating the slope (ln transformation) of an ordinary least-squares regression of the relationship between community similarity and geographic distance [34,35]. The significance of the relationship between community similarity and geographical distance or each environmental factor in each treatment was assessed by using the Mantel test [36].

Microbial community structure is shaped by a combination of deterministic processes (ecological niche-based theory) and stochastic processes (neutral theory) [37,38]. To infer the ecological factors that play a major role in the observed turnover rate between specific community assemblages, a null model approach was used to generate an expected level of βMNTD, after which the magnitude and direction of deviation between the observed βMNTD values of the test samples and the distribution of zero βMNTD values were quantified to calculate βNTI values [39,40]. Briefly, it was concluded that deterministic processes were dominant (e.g., homogeneous selection and variable selection) when |βNTI| > 2, indicating that βMNTDobs deviates from the mean βMNTD_null_ by more than two standard deviations. When |βNTI| < 2, on the other hand, stochastic factors were said to dominate (e.g., dispersal limitation, homogeneous dispersal, and undominated). To distinguish between these two cases, pairwise comparisons of the Bray–Curtis-distance-based Raup–Crick metric (RC_bray_) with |βNTI| < 2 were further calculated according to the method described by Stegen et al. [41]. That is, when |βNTI| < 2, RC_bray_ < −0.95 indicates homogeneous dispersal; |βNTI| < 2 and RC_bray_ > 0.95 represent the relative influence of dispersal limitations; and |RC_bray_| < 0.95 represents undominated processes.

## 3. Results

### 3.1. Diversity and Homogeneity of Fungal Sub-Communities in Tilled and Untilled Soils

As shown in Figure 1, there were significant differences in the alpha diversity of both abundant and rare taxa in tilled soiled compared to untilled soil. Specifically, the alpha-diversity of the Shannon Wiener index was significantly decreased in abundant taxa, and the Chao1 and Shannon Wiener indexes were significantly increased in the rare taxa in tilled soil. In contrast, the alpha-diversity of intermediate taxa was not significantly different between the two types of land management. The homogenization effect analysis, meanwhile, indicated that all the fungal sub-communities were significantly more homogenized in the tilled soil than in the untilled soil. Within that broad picture, however, the rare taxa were less homogenized than the intermediate and abundant taxa.

The Bray–Curtis-similarity-matrix-based PCo analysis revealed that the structure of the fungal sub-communities differed between the tilled and untilled soil (Figure 2), and ANOSIM analysis verified a significant variation (abundant taxa, R = 0.08, *p* = 0.003; intermediate taxa, R = 0.395, *p* = 0.001; rare taxa = 0.242, *p* = 0.001) in community composition between TT and NT treatments.

### 3.2. Niche Width and Generalists/Specialists for Abundant, Intermediate, and Rare Taxa

The niche width is an index of the biodiversity of biological utilization resources. The niche widths of the three communities were significantly different, and the niche widths of all three taxa in the tilled samples were significantly higher than those in the untilled samples (Figure 3A,B). In addition, the niche width of all abundant sub-communities was significantly higher than that of intermediate and rare taxa, which implies that abundant taxa have a better environmental adaptability than rare and intermediate taxa.

To study the differences in community structure in more depth, generalists and specialists were identified based on the niche width index (Figure 3C–H). While generalists and specialists were found in all sub-communities, the proportion of specialists in samples from tilled soil was lower than in samples from untilled soil, while, except for the rare sub-communities, the proportion of generalists was higher in the samples from tilled soil than in the samples from untilled soil.

### 3.3. Distance–Decay Patterns of Fungal Sub-Communities in Tilled and Untilled Soils

In this study, all the fungal sub-communities exhibited a significant linear distance–decay relationship in both tilled (R^2^ = 0.009, *p* = 0.002) and untilled soil treatments (R^2^ = 0.108, *p* < 0.002). Beyond that overall observation, it was evident that the distance–decay relationships of the fungal sub-communities were less pronounced in the tilled samples than in the untilled soil samples (Figure 4).

Meanwhile, both rare and intermediate taxa were observed to be more strongly subject to distance decay than abundant taxa. These findings were further confirmed by the Mantel test, and there was indeed a significant correlation (*p* < 0.05) between the similarity and spatial distance of sub-communities for all three taxa in both the tilled and untilled soils (Appendix A).

### 3.4. The Effect of Environmental Factors on Fungal Sub-Communities

The pH, TN, AP, AK, and TP increased in the tilled soil samples compared to the untilled samples, as shown in Appendix A, indicating that long-term human activities may contribute to nutrient accumulation in the soil.

The results of the Mantel test indicate that the beta-diversity (Bray–Curtis distance) of all the taxa in the tilled soil samples showed a higher correlation between environmental factors than in the untilled samples (Figure 5). In contrast, no correlation was found between environmental factors and beta-diversity in the untilled samples. Moreover, in the tilled samples, rare and intermediate taxa were significantly correlated with more environmental factors than abundant taxa. In detail, abundant taxa were only significantly correlated with SOM, while intermediate taxa were significantly correlated with TN, SOM, and AK, and rare taxa were significantly correlated with pH, TN, SOM, AK, and TK. This implies that environmental factors have a greater ability to shape rare and intermediate taxa than abundant taxa.

### 3.5. Assembly Processes in Abundant, Intermediate, and Rare Fungal Sub-Communities in the Tilled and Untilled Soils

In both tilled and untilled soil, stochastic processes contributed more than 88% of the assembly processes for abundant fungal taxa, while deterministic processes accounted for more than 55% of the processes shaping intermediate and rare sub-communities (Figure 6A). This suggests that deterministic processes play an important role in the construction of rare and intermediate fungal sub-communities, while stochastic processes play a major role in shaping abundant fungal sub-communities. Meanwhile, we also observed that the community assembly process of the three taxa varied consistently in the tilled versus the untilled soil, being more determined in the latter.

Quantitative analysis was also used to explain the assembly process of fungal taxa more fully (Figure 6B). For the abundant sub-community, dispersal limitation contributed a large fraction to the assembly in both tilled and untilled samples, followed by variable selection. On the contrary, variable selection contributed a larger fraction to the assembly than dispersal limitation in both tilled and untilled samples. In addition, for all fungal taxa, the proportion of variable selection in untilled samples was greater than that in tilled samples.

## 4. Discussion

### 4.1. Distinct Fungal Sub-Community Structures between Tilled and Untilled Soils

This study investigated the composition and diversity of fungal sub-communities under different land management practices. Our results showed both that the structure of fungal taxa differed between tilled and untilled samples and that the Chao1 and Shannon indexes increased significantly in the rare taxa. Our results also showed that the abundant, intermediate, and rare microbial sub-communities were all more homogenous in tilled than in untilled soil. This finding contrasts with those in many studies and is significant since a more homogeneous microbial community is not conducive to biodiversity conservation [10]. This finding may be due to a decrease in specialized species and an increase in generalized species in the community [9], since such a trend has previously been shown to lead to homogenization [42]. In other words, the intensive land management characterized by tilled soil leads to homogenization across the whole range of taxa in the microbial community. Within that big picture, however, it is very interesting that the diversity of rare taxa actually increased significantly in tilled soil. This is surprising since, in general, it would be expected that rare taxa would be more likely to experience a decrease in diversity since they are generally seen as sensitive to external changes. Previous work, for example, has shown that many of the community functions typically provided by rare taxa are lost in intensive land management [43,44]. An explanation, however, might lie in the observation that increased crop diversity has a positive impact on soil microbial abundance and diversity. From that perspective, the increase in the diversity of rare taxa might be due to the change in the soil micro-environment caused by increased crop diversity [45,46]. Tillage is known to increase surface cover and soil nutrient content, mainly by the accumulation of root secretions and plant apoplast left over from previous crops [47], and these changes can create favorable conditions for the growth of some soil microorganisms, and stimulate the growth of specific microbial communities latent in the soil [46,48]. For example, the content and diversity of plant litter are known to increase in tilled soil samples, which may lead to an increase in the diversity of microorganisms involved in decomposition [49,50]. This may be one reason for the increase in rare taxa in the tilled treatments. Meanwhile, in this study, we did not consider the individual specificity of intensive management. Compared to large-scale, intensively managed plots, the plots sampled in this experiment were all small-scale and intensively managed, with a high degree of individual specificity. For example, the variety of crops grown, the type of fertilizer applied, and the tillage practices may vary among individuals, and the consistency of variation in these factors versus large-scale intensive management is lower, which may affect the soil environment and directly or indirectly affect the microbial community. In other words, compared to large-scale intensive management, small-scale intensive management, i.e., management with individual specificity, may be beneficial for the conservation of microbial diversity.

### 4.2. Fungal Sub-Communities in Tilled and Untilled Soils Exhibited Different Biogeographic Patterns

Distance–decay relationships were used to analyze the variation between community similarity and geographic distance [34]. The slope and R^2^ in this method were applied to investigate the rate and intensity of community decay [51]. Our results reveal a significant distance–decay relationship (*p* < 0.001) for all three taxa in both tilled and untilled systems, but the observed distance–decay relationship was weaker in both systems. Although many studies have found distance–decay relationships of different strengths in microbial communities, other studies have found little evidence of such relationships [52,53], a difference that may be due to spatial range size and habitat differences [32,54]. Generally, spatial extent is positively correlated with the strength of distance–decay relationships because at large spatial scales [55,56], the community similarity reduces, leading to stronger distance–decay relationships [57,58]. This effect may be cloaked in our study due to the relatively small spatial extent of our sample. Alternatively, it is possible that the distance–decay relationship of microbes may vary depending on the different environmental or ecological contexts under study. For example, aqueous systems are supposed to be weaker than others due to their higher dispersal capacity, and, in soil systems, the distance–decay relationship is weaker in deeper soil layers [59]. In addition, the slope of the distance decay was steeper in the untilled samples in our study than in the tilled samples, which coincided with the results of the Mantel test, indicating that geographic separation had a greater effect on the untilled samples. This suggests that tillage weakened the effect of the distance–decay relationship.

Reducing microbial beta-diversity by increasing the proportion of shared species between distant communities increases the homogenization effect of the community and thus reduces the distance–decay relationship [60]. This pattern was also present in our study, where the proportion of specialized species decreased and the proportion of generalized species increased in the tilled samples, while the homogenization intensity of all taxa was significantly higher in the tilled samples than in the untilled ones, which in turn led to a decrease in the distance–decay relationship. This also indicates that, for tilled samples, the homogenization effect leads to a weakening of distance decay, and the similarity in the spatial distance of fungal sub-communities in the tilled samples is greater than that in the untilled samples. In contrast, fungal sub-communities were more geographically specific in the untilled samples. Different distance–decay relationships were found for abundant, intermediate, and rare taxa in both our tilled and untilled samples, with a stronger distance–decay relationship for rare and intermediate taxa than for abundant taxa, possibly due to the fact that intermediate and rare taxa are more heterogeneous than abundant taxa, as well as relevant to the environmental adaptability and community cover of abundant, intermediate, and rare taxa [58,61].

In addition, previous work has shown that whether or not soil is tilled has a significant impact on soil nutrient parameters, and these differences will also have a direct or indirect impact on microbial communities [62,63]. In our study, we observed that TN, AP, AK, and TP increased significantly in the tilled samples, while the three fungal taxa showed significant correlations with environmental factors. Geographic distance, meanwhile, had a smaller influence on the composition of the fungal sub-community in the tilled soil than that of the untilled, possibly owing to long-term human activities, such as cropping patterns and environmental factors, that diminish the contribution of geographic distance to the tilled sample [64].

### 4.3. Relative Contributions of Stochastic and Deterministic Processes in Tilled and Untilled Soils

Based on a null model approach, the data were fitted to calculate the relative roles of stochastic and deterministic processes for the fungal sub-community. The results indicate that although both stochastic and deterministic processes have important roles, they have different contributions to fungal sub-community construction in different treatments. It is generally assumed that microbial communities tend to evolve toward deterministic processes when external selection by the environment is present [65,66]. It is interesting to note, however, that, for all three taxa in our study, the community assembly process evolved toward a stochastic process after the application of selection pressure. A possible reason for this phenomenon is that tillage increases the diversity of fungal sub-communities, and high microbial diversity leads to the development of stochastic processes [67]. The higher community diversity of rare and intermediate taxa encouraged a shift toward stochastic processes of community assembly. However, the variation in diversity of abundant taxa showed opposite results to those of rare and intermediate taxa, but the assembly process realized a similar trend, probably related to the high environmental adaptability of abundant taxa [44,68].

It is considered that species with wider ecological niche widths are generalists, and, due to a higher environmental tolerance, that these are less affected by environmental factors [69]. In our study, the niche widths of both rare and intermediate sub-communities of fungi were higher in the tilled treatments than in the untilled, and it is known that fungal sub-communities with higher niche widths can efficiently use a range of resources and are therefore less subject to environmental filtering [31]. Thus, rare and intermediate fungal sub-communities should be less subject to deterministic processes in tilled compared to untilled soils. Our results confirm that the assembly process of rare taxa, both in tilled and untilled soil, was more deterministic relative to that of abundant taxa. Rare microbial sub-communities were more environmentally influenced than abundant microbial sub-communities, with a smaller niche width and generally lower environmental adaptability [70], while abundant species could occupy a wider nice width and exhibit a capability to adapt to wider environmental gradients [16]. Thus, abundant microbial taxa have a greater environmental adaptability and greater stochasticity than rare microbial taxa [71].

According to RC_bray_ analysis, dispersal limitation and variable selection are the main ecological processes that shape microbial communities. It has been suggested that abundant microbial taxa are more restricted by dispersal than rare taxa in both aquatic and terrestrial ecosystems [23,32,72], and that contention is supported by our results, with abundant sub-communities being more affected by dispersal limitation than rare sub-communities. Tillage, meanwhile, reduced the importance of variable selection in community construction. The enrichment of soil nutrients by tillage treatment would reduce the effect of variable selection on community construction [73]. The decrease in specialized species in the three taxa may explain that decrease in variable selection in tilled soils [74]. Taxa selected in one region may not be selected in another region given regional differences in selection environments [75]. The colonization of specific taxa may create unique niches, and in such cases will result in local differences in community composition, thus increasing the relative importance of variable selection [74,76].

## 5. Conclusions

In this study, we found that the response of fungal sub-communities varied under different land management practices. Increased anthropogenic disturbances associated with tillage reduced the alpha-diversity of the abundant fungal sub-community but increased that of the rare fungal sub-community. It also changed the composition of the fungal community, reducing the proportion of specialized species in the community. The homogenization of all three fungal sub-communities was also stronger in anthropogenically disturbed tilled lands than in untilled lands, which further reduced the distance–decay relationship of fungal sub-communities. In comparison, environmental variables play a dominant role in shaping fungal communities in tilled soils, while geographical factors play a major role in untilled soils. In addition, the ecological niche occupied by fungi became wider after tilled treatment, which resulted in the construction of fungal communities coming to be influenced less by deterministic factors and more by stochastic ones. Overall, these results deepen the understanding of the response of different soil fungal sub-communities to anthropogenic disturbances and provide some insights into fungal changes under different land management practices.

## Figures and Tables

**Figure 1 jof-09-00646-f001:**
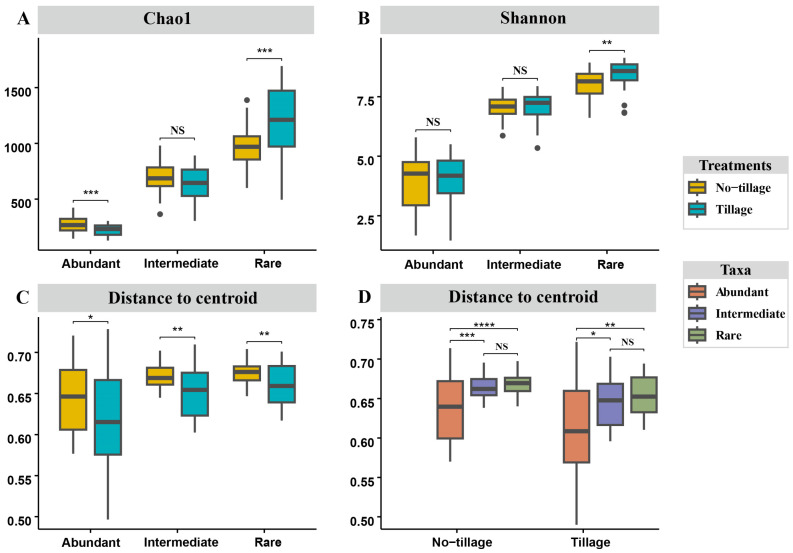
Diversity and homogeneity of soil fungal sub-communities. (**A**) Chao1 and (**B**) Shannon indexes of fungal sub-communities under different land management measures; Homogeneity of soil fungal sub-communities under different land management measures (**C**) and under the same land management measures (**D**). Significant differences were tested by Wilcoxon test (**** *p* < 0.0001, *** *p* < 0.001, ** *p* < 0.01, * *p* < 0.05, NS: No significant).

**Figure 2 jof-09-00646-f002:**
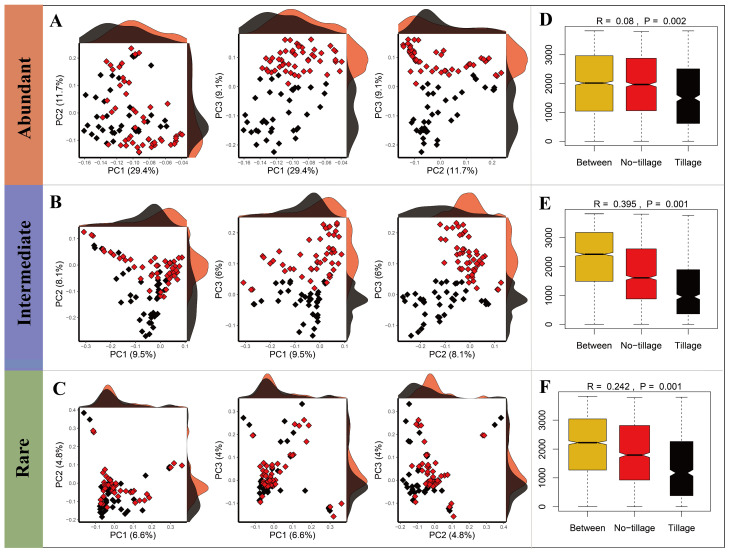
Differential analysis of the composition of three sub-communities in tilled and untilled lands. PCo analysis for abundant (**A**), intermediate (**B**), and rare (**C)** sub-communities. Analysis of similarities (ANOSIM) for abundant (**D**), intermediate (**E**), and rare (**F**) sub-communities. The red and black plots represent untilled and tilled soils, respectively.

**Figure 3 jof-09-00646-f003:**
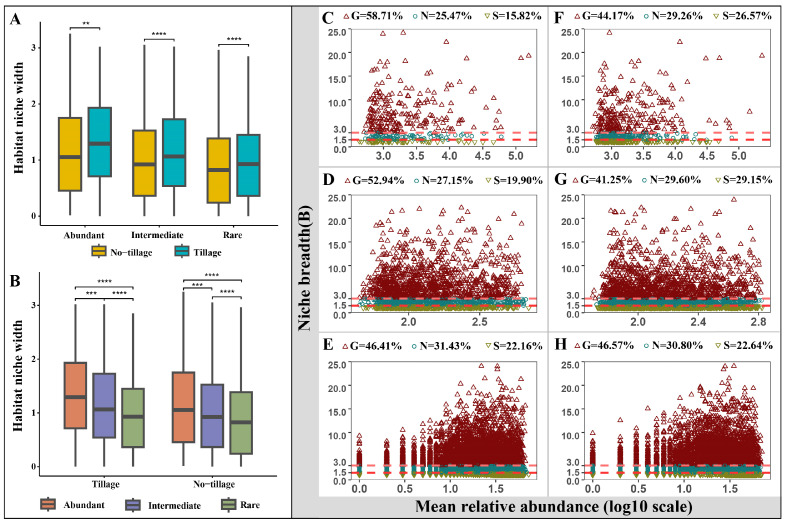
Niche breadth and specialization/generalization of soil fungal sub-communities. Niche breadth of soil fungal sub-communities under different land management measures (**A**) and under the same land management measures (**B**); (**C**–**H**) Proportion of specialized/generalized species in the fungal sub-communities under different land management measures. Abundant sub-community in tilled soil (**C**) and untilled soil (**F**); intermediate sub-community in tilled soil (**D**) and untilled soil (**G**); and rare sub-community in tilled soil (**E**) and untilled soil (**H**). Significant differences were tested by the Wilcoxon test (**** *p* < 0.0001, *** *p* < 0.001, ** *p* < 0.01).

**Figure 4 jof-09-00646-f004:**
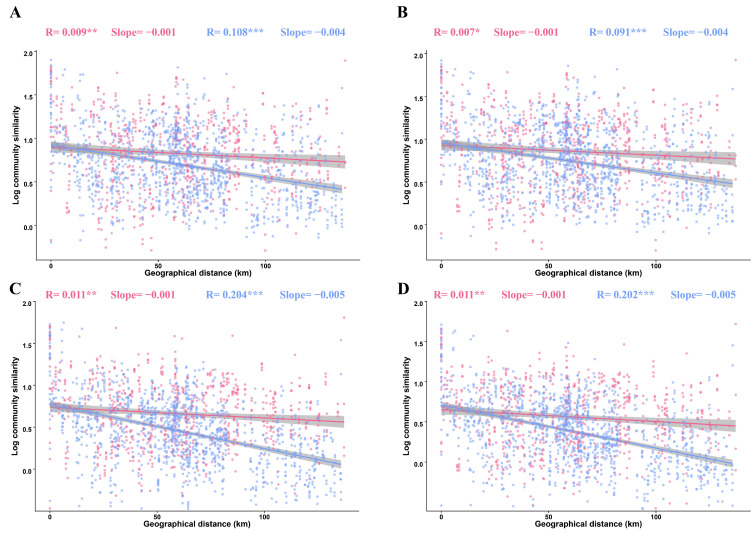
The distance–decay relationships of fungal sub-communities. (**A**) Fungal communities under different land management measures; (**B**) abundant taxa, (**C**) intermediate taxa, and (**D**) rare taxa sub-communities under different land management measures. The red and blue lines represent tilled and untilled soils, respectively. (*** *p* < 0.001, ** *p* < 0.01, * *p* < 0.05).

**Figure 5 jof-09-00646-f005:**
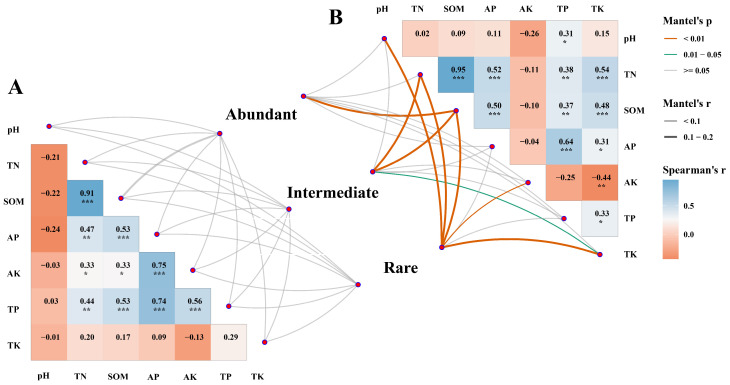
Pairwise correlations between environmental variables, and the Mantel tests between three fungal sub-communities’ similarities vs. the environmental factors. Three untilled taxa (**A**) and three tilled taxa (**B**). The lines represent significant relationships, where the width of the line represents the Mantel r statistic value and the different colors of the lines represent different degrees of significance. The Spearman correlation coefficient between environmental variables is shown in the heat map matrix. (*** *p* < 0.001, ** *p* < 0.01, * *p* < 0.05).

**Figure 6 jof-09-00646-f006:**
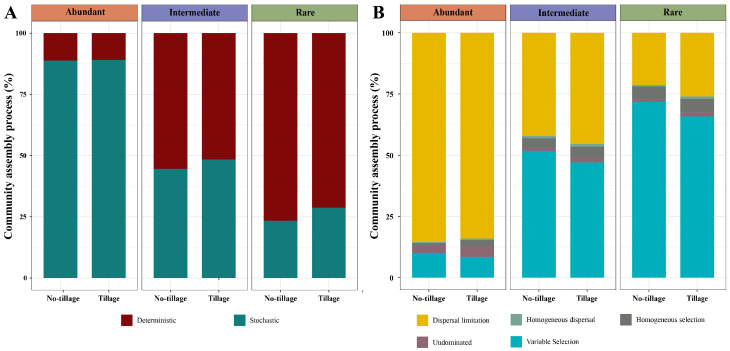
Assembly processes of fungal sub-community under tilled and untilled land management. (**A**) The proportion of deterministic or stochastic processes in the three taxa; (**B**) Quantitative calculation of community assembly process.

## Data Availability

All data supporting the findings of this study are available on request from the corresponding authors (Hui Cao).

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
