# Peer review of "Biogeographic Patterns of Fungal Sub-Communities under Different Land-Use Types in Subtropical China"

_jof, 2023, doi:10.3390/jof9060646_

Round 1
Reviewer 1 Report
The manuscript is original, well written and organised and of interest. I have only few comments:
In line 28 I suggest to focus more on ecosystem functionality rather than stability.
Lines 58-60: the sentence is not clear, please revise it.
Figure 4: I suggest to change colours to enhance readability.
Author Response
The manuscript is original, well written and organised and of interest. I have only few comments:
Comment 1: In line 28 I suggest to focus more on ecosystem functionality rather than stability.
Answer: Thank you for pointing that out. We made correction in revised manuscript.
Comment 2:Lines 58-60: the sentence is not clear, please revise it.
Answer: We are very sorry for our poor presentation, and we have rewritten this section in the revised version.
Comment 3: Figure 4: I suggest to change colours to enhance readability.
Answer: Thank you very much for your constructive comments. In the revised manuscript, we have made changes to the colors of the figure.

Reviewer 2 Report
The manuscript presented to a review is written correctly. It offers a high substantive level. As for the impact of use, it has been studied in forestry. This work focuses on agriculture. It is good work.
Author Response
Comment: The manuscript presented to a review is written correctly. It offers a high substantive level. As for the impact of use, it has been studied in forestry. This work focuses on agriculture. It is good work.
Answer:Thank you very much for your recognition and positive feedback on our manuscript. We greatly appreciate your time and effort in thoroughly reviewing our work. Once again, thank you very much for your high appreciation of our work. We will continue to strive to improve the quality of our work.

Reviewer 3 Report
The manuscript "Biogeographic patterns of fungal sub-communities under different land-use types in subtropical China" provides an excellent meta-genomic comparison of the soil communities. The statistical methods are well applied, and there is substantial scientific insight. Please see some minor comments to make the results more readable:
1) Figures 1, 2, 3, 4, and 5 - Please provide the complete spelling of the acronyms in the legend. AT MT, RT, NN, TT. This will make the plots self-explanatory, and the reader doesn't need to go back to the text
2) Figures 1, 3, and 4 - Please provide a full p-value significance in the legend. Only two levels of p-values are described in the legend, but the plots have several *, **, and ***. What are their thresholds? Please explain it better in the legend section
2b) Figure 2 - Please use the same colour code as in Figures 1 and 3.. for "Rare", "Intermediate", and "Abundant" labels (A, B and C plots). The colours of these three labels should all be consistent across the manuscript.
3) Figure 5 - Some correlation values must be more readable in the boxes. Dark blue, especially. Please make the blue squares less dark (change the colour scale).
4) Figure 2 D E F - What are the colours for TT and NT? Why green (blue), red, and yellow (between)? You can use black and red here for consistency with the other figures
5) Figure 4 - Please use transparency (alpha parameter in ggplot). This will help you show the overlapping dots in the plot. So far, it's difficult to see the dots well because they are too many, especially between the two colours (the blue ones cover red dots)
6) Please make the genomic reads public by sharing them on NCBI-SRA. It's important that everyone can use your data! Thank you
7) In the intro, you mention this "Recent studies, however, have shown that rare fungal taxa, which likewise include diverse genes and functions...". Is there any concrete example you found from your analysis that you can add about genes and function? Your manuscript deals with meta-analysis, but could you add anything more about gene function? This is more a curiosity for new and probably an interesting analysis for future publications
Author Response
The manuscript "Biogeographic patterns of fungal sub-communities under different land-use types in subtropical China" provides an excellent meta-genomic comparison of the soil communities. The statistical methods are well applied, and there is substantial scientific insight. Please see some minor comments to make the results more readable:
Comment 1: Figures 1, 2, 3, 4, and 5 - Please provide the complete spelling of the acronyms in the legend. AT MT, RT, NN, TT. This will make the plots self-explanatory, and the reader doesn't need to go back to the text.
Answer: Thank you very much for your constructive comments. The legend of these figures of the manuscript has been revised.
Comment 2: Figures 1, 3, and 4 - Please provide a full p-value significance in the legend. Only two levels of p-values are described in the legend, but the plots have several *, **, and ***. What are their thresholds? Please explain it better in the legend section.
Answer: We are very sorry for our poor presentation. The threshold values are ****p < 0.0001, ***p < 0.001, **p < 0.01, *p < 0.05.In the revised manuscript, we explain this.
Comment 3: Figure 2 - Please use the same colour code as in Figures 1 and 3.. for "Rare", "Intermediate", and "Abundant" labels (A, B and C plots). The colours of these three labels should all be consistent across the manuscript.
Answer: We are very sorry for our poor presentation. In the revised version, we have made changes to these colors to keep them consistent.
Comment 4: Figure 5 - Some correlation values must be more readable in the boxes. Dark blue, especially. Please make the blue squares less dark (change the colour scale).
Answer: We are very sorry for our poor presentation. With reference to your comments, we have made adjustments to the colors so that they can be displayed more clearly.
Comment 5: Figure 2 D E F - What are the colours for TT and NT? Why green (blue), red, and yellow (between)? You can use black and red here for consistency with the other figures.
Answer: We are very sorry for our poor presentation. The colors of TT and TN in the original figure are red and blue, respectively. With reference to your comments, we have modified them to be consistent.
Comment 6: Figure 4 - Please use transparency (alpha parameter in ggplot). This will help you show the overlapping dots in the plot. So far, it's difficult to see the dots well because they are too many, especially between the two colours (the blue ones cover red dots).
Answer: We are very sorry for our poor presentation. We have modified the transparency of the dot so that it can be displayed clearly.
Comment 7: Please make the genomic reads public by sharing them on NCBI-SRA. It's important that everyone can use your data! Thank you
Answer: Thank you very much for your suggestion. Genomic sequencing data used in this study were submitted to the NCBI Sequence Read Archive (SRA) under the accession number SRR24744258 to SRR24744345.
Comment 8: In the intro, you mention this "Recent studies, however, have shown that rare fungal taxa, which likewise include diverse genes and functions...". Is there any concrete example you found from your analysis that you can add about genes and function? Your manuscript deals with meta-analysis, but could you add anything more about gene function? This is more a curiosity for new and probably an interesting analysis for future publications.
Answer: Thank you very much for providing insightful suggestions. But we are very sorry, our current research does not involve gene functional aspects. Regarding the aspect of functionality, we have considered treating it as a separate topic for detailed discussion in future manuscript. Our subsequent research will focus on revealing changes in functionality through the integration of bacterial and fungal taxa along with specific functional gene groups. The main objective of this study is to analyze and compare the variations in soil fungal sub-communities under different land management practices. The mentioned content in the introduction ("Recent studies, however, have shown that rare fungal taxa, which likewise include diverse genes and functions...") primarily emphasizes the unique characteristics and commonalities among different fungal sub-communities, highlighting the importance of distinguishing fungal communities into distinct sub-communities. However, most studies tend to overlook this aspect and treat fungal communities as a whole, whereas our study specifically emphasizes this aspect. We aim to investigate the responses of different fungal sub-communities to anthropogenic disturbances, focusing on a relatively large spatial scale. Our focus is on exploring the impacts of different land management practices on fungal sub-communities diversity, environmental adaptability, distance decay relationships, and assembly processes. Regarding the research on community functionality, the current studies mainly rely on the evaluation of functional gene groups to reveal the functional characteristics of microbial communities. Merely relying on functional predictions for functional analysis may not provide a comprehensive understanding in the context of this study, and it is not aligned with the current focus of our research. Therefore, we believe that treating functionality as a separate topic would better ensure the integrity and independence of both research aspects. Once again, we appreciate your suggestions and look forward to future collaborations.
